# Doxazosin, a Classic Alpha 1-Adrenoceptor Antagonist, Overcomes Osimertinib Resistance in Cancer Cells via the Upregulation of Autophagy as Drug Repurposing

**DOI:** 10.3390/biomedicines8080273

**Published:** 2020-08-05

**Authors:** Shuhei Suzuki, Masahiro Yamamoto, Tomomi Sanomachi, Keita Togashi, Asuka Sugai, Shizuka Seino, Masashi Okada, Takashi Yoshioka, Chifumi Kitanaka

**Affiliations:** 1Department of Molecular Cancer Science, Yamagata University School of Medicine, 2-2-2 Iida-nishi, Yamagata 990-9585, Japan; t-sanomachi@med.id.yamagata-u.ac.jp (T.S.); ke-togashi@med.id.yamagata-u.ac.jp (K.T.); s-asuka@med.id.yamagata-u.ac.jp (A.S.); s.sizuka@med.id.yamagata-u.ac.jp (S.S); m-okada@med.id.yamagata-u.ac.jp (M.O.); ckitanak@med.id.yamagata-u.ac.jp (C.K.); 2Department of Clinical Oncology, Yamagata University School of Medicine, 2-2-2 Iida-nishi, Yamagata 990-9585, Japan; ytakashi@med.id.yamagata-u.ac.jp; 3Department of Ophthalmology and Visual Sciences, Yamagata University School of Medicine, 2-2-2 Iida-nishi, Yamagata 990-9585, Japan; 4Research Institute for Promotion of Medical Sciences, Yamagata University Faculty of Medicine, 2-2-2 Iida-nishi, Yamagata 990-9585, Japan

**Keywords:** drug resistance, doxazosin, osimertinib, autophagy, drug repositioning, drug repurposing, cancer stem cells

## Abstract

Osimertinib, which is a third-generation epidermal growth factor receptor tyrosine kinase inhibitor, is an important anticancer drug because of its high efficacy and excellent safety profile. However, resistance against osimertinib is inevitable; therefore, therapeutic strategies to overcome the resistance are needed. Doxazosin, a classic quinazoline-based alpha 1-adrenoceptor antagonist is used to treat hypertension and benign prostatic hyperplasia with a known safety profile. The anticancer effects of doxazosin have been examined in various types of malignancies from the viewpoint of drug repositioning or repurposing. However, it currently remains unclear whether doxazosin sensitizes cancer cells to osimertinib. Herein, we demonstrated that doxazosin induced autophagy and enhanced the anticancer effects of osimertinib on the cancer cells and cancer stem cells of non-small cell lung cancer, pancreatic cancer, and glioblastoma at a concentration at which the growth of non-tumor cells was not affected. The osimertinib-sensitizing effects of doxazosin were suppressed by 3-methyladenine, an inhibitor of autophagy, which suggested that the effects of doxazosin were mediated by autophagy. The present study provides evidence for the efficacy of doxazosin as a combination therapy with osimertinib to overcome resistance against osimertinib.

## 1. Introduction

Osimertinib, which is a third-generation epidermal growth factor tyrosine kinase inhibitor (EGFR-TKI), is an important drug in the treatment of cancer because of its strong anticancer activities and mild adverse effects [1,2]. Osimertinib is used to treat non-small cell lung cancer (NSCLC) harboring *EGFR* gene mutations, such as the deletion of exon 19 and L858R mutations [1,2]. However, acquired resistance against osimertinib inevitably occurs through several mechanisms which include the acquisition of C797S mutations, small cell transformation, and the activation of bypass signal pathways, such as *HER2* amplification, *MET* amplification, and *PIK3CA* activation mutations [3,4,5]. Moreover, *EGFR* wild-type NSCLC, accounting for approximately 80–85% of cases, are primarily resistant to osimertinib [6,7]. Although EGFR signaling is dysregulated in other types of malignancies, including pancreatic cancer and glioblastoma [8,9], EGFR-TKI exhibits limited efficacy against these malignancies [10,11]. Many studies, including ours, have revealed potential targets to overcome EGFR-TKI resistance, including the downregulation of survivin [12], blockade of glucose uptake [13], inhibition of YAP [14], and modulation of autophagy [15,16,17,18]. However, these treatment strategies have not yet been applied to clinical settings. 

Doxazosin is a quinazoline-based alpha 1-adrenoceptor antagonist which is used in the treatment of hypertension [19] and benign prostatic hyperplasia [20]. Drug repositioning or repurposing is a strategy that utilizes drugs that have already been approved for new indications, thereby reducing the time and cost associated with the development of new drugs because of their known dose and toxicity profiles [21]. For example, thalidomide, which was developed as an anti-emetic drug, is used in the treatment of multiple myeloma [22]. The anticancer effects of doxazosin have been reported in several types of cancer cell lines, including prostate cancer, renal cell carcinoma, urothelial carcinoma, pancreatic cancer, glioblastoma, breast cancer, ovarian cancer, colon cancer, stomach cancer, and thyroid cancer [23]. The anticancer effects of doxazosin are mediated through several mechanisms including the activation of TGFβ and IκB [24,25], downregulation of DNA replication/repair genes [26,27], inhibition of PKB/AKT activation [28], inhibition of angiogenesis [29], and induction of autophagy [30,31]. However, it currently remains unclear whether doxazosin enhances the anticancer effects of osimertinib in cancer cells.

In the present study, we revealed that doxazosin induced autophagy and sensitized various types of cancer cells and cancer stem cells (CSCs) to osimertinib.

## 2. Materials and Methods 

### 2.1. Antibodies and Reagents

Anti-LC3 (#3868) and anti-cleaved caspase 3 (#9661) antibodies were purchased from Cell Signaling Technology, Inc. (Beverly, MA, USA). The anti-SQSTM1 (p62, sc-28359) antibody was purchased from Santa Cruz Biotechnology (Santa Cruz, CA, USA). The anti-β-actin (A1978) antibody was from Sigma (St. Louis, MO, USA). Osimertinib was purchased from Chemscene LLC. (Monmouth Junction, NJ, USA) and dissolved in dimethyl sulfoxide (DMSO) to 10 mM as a stock solution. Doxazosin was from Tokyo Chemical Industry Co., Ltd. (Tokyo, Japan) and dissolved in DMSO to prepare a 10 mM stock solution. 3-Methyladenine (3-MA) was from Merck Millipore (Darmstadt, Germany) and dissolved in DMSO to 75 mM as a stock solution. Bafilomycin A1 was from Sigma and dissolved in DMSO to 100 µM as a stock solution.

### 2.2. Cell Cultures

The A549 and PC-9 human NSCLC cell lines were obtained from the Riken BioResource Center (Tsukuba, Japan). The PANC-1 human pancreatic cancer cell line was from the Cell Resource Center for Biomedical Research, Institute of Development, Aging and Cancer, Tohoku University (Sendai, Japan). The A549 and PANC-1 cells were cultured in DMEM/F12 medium, and the PC-9 cells in RPMI1640 medium. These cell culture media were supplemented with 10% fetal bovine serum (FBS), 100 units/mL of penicillin, and 100 μg/mL of streptomycin. The PC-9-OR osimertinib-resistant PC-9 cells were established by culturing in increasing concentrations of osimertinib (0.1–1.5 μM) over a two-month period, as previously described [32]. The PC-9-OR cells were maintained in the culture media containing 1.5 μM osimertinib. The establishment of CSC lines from A549 and PANC-1 (A549 CSLC and PANC-1 CSLC, respectively) was previously reported [33,34]. The authenticity of the A549 CSLC and PANC-1 CSLC lines was verified by genotypes of short tandem repeat (STR) loci (Bio-Synthesis Inc., Lewisville, TX, USA) and comparisons with the ATCC STR database for Human Cell Lines. GS-Y01 is a CSC line established from a glioblastoma patient [35]. These CSCs were cultured as previously described [34,36]. In brief, these CSCs were cultured on collagen I-coated dishes (IWAKI, Tokyo, Japan) in stem cell culture medium (DMEM/F12 medium added with 1% B27 supplement (Gibco-BRL, Carlsbad, CA, USA), 20 ng/mL of EGF and FGF2 (PeproTech Inc., Rocky Hill, NJ, USA), D-(+)-glucose (final concentration, 26.2 mM), L-glutamine (final concentration, 4.5 mM), 100 units/mL of penicillin, and 100 μg/mL of streptomycin). Stem cell culture medium was changed every 3 days, and EGF and FGF2 were added every day. GS-Y01 cells were induced to differentiate by culturing in DMEM/F12 medium with 10% FBS and antibiotics, as described above, for 14 days. IMR-90, normal human fetal lung fibroblasts, were purchased from the American Type Culture Collection and cultured in DMEM/F12 supplemented with 10% FBS, 100 units/mL penicillin, and 100 μg/mL streptomycin. All experiments with IMR-90 were performed within a low passage number (less than eight). 

### 2.3. Cell Viability Assay

Viable and dead cells were both identified by their ability and inability, respectively, to exclude vital dyes [37,38]. In brief, all cells were stained with 0.2% trypan blue, and the numbers of viable and dead cells were counted using a hemocytometer. Dead cells (%) were defined as 100 × the number of dead cells / (the number of viable cells + the number of dead cells). The growth inhibition rate (%) was defined as 100 × (1 - the number of viable cells treated with osimertinib and / or doxazosin / the number of viable cells without treatment).

### 2.4. Immunoblot Analysis

Cells were harvested, washed with PBS, and then lysed in RIPA buffer (10 mM Tris-HCl (pH 7.4), 0.1% SDS, 0.1% sodium deoxycholate, 1% NP-40, 150 mM NaCl, 1 mM EDTA, 1.5 mM Na_3_VO_4_, 10 mM NaF, 10 mM sodium pyrophosphate, 10 mM sodium β-glycerophosphate, and 1% protease inhibitor cocktail set III Sigma). After centrifugation at 14,000 × *g* at 4 °C for 10 min, supernatants were saved as cell lysate samples, and the protein concentrations of cell lysates were measured using a BCA protein assay kit (Pierce Biotechnology Inc., Rockford, IL, USA). Cell lysates containing equal amounts of protein were separated by SDS-PAGE and transferred to polyvinylidene difluoride membranes. Membranes were probed with primary antibodies (1:1500 dilution for all primary antibodies), and then with an appropriate HRP-conjugated secondary antibody (Jackson ImmunoResearch Laboratories, West Grove, PA, USA), according to the manufacturer’s instructions. Immunoreactive bands were visualized using Immobilon Western Chemiluminescent HRP Substrate (Merck Millipore). The relative density of immunoreactive bands was analyzed by densitometry using ImageJ 1.52a software (National Institutes of Health, Bethesda, MD, USA).

### 2.5. Immunofluorescence Analysis

The protocol of the immunofluorescence analysis was modified from a previous study [39]. In brief, A549 cells were seeded on coverslips in 35 mm dishes and used in experiments. After cells were fixed with 4% (*w/v*) paraformaldehyde at room temperature (RT) for 10 min and washed with PBS three times, cells were permeabilized and blocked with 0.4% Triton X-100/2% FBS in PBS at RT, for 15 min. After being washed with PBS three times, cells were incubated with a primary antibody in PBS containing 2% FBS at RT, for 60 min, and then incubated with Alexa Fluor 488-conjugated secondary antibody (A11034, Thermo Fisher Scientific, San Jose, CA, USA) and Hoechst 33342 (10 μg/mL) in the same buffer at RT for 30 min. Fluorescent images were acquired using a confocal laser-scanning microscope (FLUOVIEW FV10i OLYMPUS, Tokyo, Japan).

### 2.6. Statistical Analysis

Results are expressed as means and standard deviations (SD). Differences were compared using the two-tailed Student’s *t*-test. P-values < 0.05 were considered to be significant and were indicated with asterisks.

## 3. Results

### 3.1. Doxazosin Inhibits Cancer Cell Growth and Is Cytotoxic to Cancer Cells and Cancer Stem Cells, but not to Non-Cancer Cells

We examined whether doxazosin exerts anticancer effects on the cancer cells of NSCLC, pancreatic cancer, and glioblastoma. We treated NSCLC and pancreatic cancer cell lines (A549 and PANC-1, respectively), glioblastoma cells (GS-Y01 glioma stem cells whose differentiation was induced by a treatment with medium supplemented with 10% FBS for ~14 days, differentiated GS-Y01), and one subline (PC-9-OR, PC-9-osimertinib resistant) with doxazosin, and then subjected them to a cell viability assay. The doxazosin treatment inhibited cancer cell growth and induced cancer cell death in a dose-dependent manner (Figure 1a). To identify the nontoxic dose of doxazosin in non-cancer cells, we treated IMR-90 cells, non-cancer human fibroblasts, with doxazosin at the same concentration as cancer cells. Doxazosin was not toxic to normal cells, except at the highest concentration (50 µM) (Figure 1b).

Since it currently remains unknown whether doxazosin exerts anticancer effects on CSCs, we investigated if doxazosin exerts inhibitory effects on CSCs derived from A549 and PANC-1 cells (A549 CSLC and PANC-1 CSLC, respectively) and GS-Y01 cells. Properties of these cell lines as cancer stem cells were validated previously [37,40]. Doxazosin induced cell death and inhibited the growth of CSCs in a dose-dependent manner (Appendix A). These results suggest that doxazosin, at an intermediate concentration (10–25 µM), is not toxic to normal cells and induces cytotoxic effects in a cancer cell-specific manner.

### 3.2. Doxazosin Enhances Anticancer Effects of Osimertinib in Cancer Cells and CSCs

We examined whether doxazosin sensitizes cancer cells to osimertinib, a third-generation EGFR-TKI. We treated A549, PANC-1, PC-9-OR, and differentiated GS-Y01 cells with osimertinib in combination with doxazosin. We chose the concentrations of 2 and 15 µM for osimertinib and doxazosin, respectively, because the concentrations of the drugs suppress the viable cell number by approximately 10–40% in all the cell lines. The combination of osimertinib with doxazosin suppressed cell growth and induced cell death significantly more than a treatment with either drug alone (Figure 2). Similarly, the combination suppressed cell growth and induced cell death more than the treatment with either drug in A549 CSLC, PANC-1 CSLC, and GS-Y01 CSCs (Appendix A). These results indicated that doxazosin enhanced the anticancer effects of osimertinib in cancer cells and CSCs.

### 3.3. Doxazosin Induces the Activation of Autophagy

Since alpha 1-adrenoceptor antagonists have been reported to induce autophagy in prostate cancer cells [30,31] and autophagy plays a role in resistance against EGFR-TKI [15,16,17,18], we examined if osimertinib and doxazosin induced autophagy in cancer cells and CSCs. Although osimertinib has previously been shown to induce autophagy in NSCLC cells [41,42], the treatment with osimertinib did not apparently affect the levels of LC3-II in A549 cells (Figure 3a), PANC-1, A549 CSLC, and PANC-1 CSLC cells (Appendix A). The doxazosin treatment increased LC3-II and p62 levels in cancer cells (Figure 3b), and these changes were accompanied by an increased number of LC3 puncta (Figure 3c). Furthermore, the treatment of bafilomycin A1, an inhibitor of autophagosome-lysosome fusion, enhanced the level of LC3-II induced by doxazosin (Figure 3d). Because activation of p62 is involved in selective autophagy [43], the results (p62 increase induced by doxazosin) possibly suggest that autophagy mediated by doxazosin could be selective autophagy. Doxazosin also increased LC3-II levels in CSCs (Appendix A). The combination of osimertinib with doxazosin did not appear to significantly affect doxazosin-induced autophagy (Appendix A). Furthermore, because it is reported that doxazosin induces apoptosis [44,45], we investigated the involvement of apoptotic cell death in the chemosensitization effect of doxazosin by analyzing the levels of cleaved caspase 3. However, doxazosin, osimertinib, and their combination did not consistently induce apoptosis in A549, PANC-1, A549 CSLC, and PANC-1 CSLC (Appendix A).

### 3.4. The Autophagy Inhibitor, 3-Methyladenine, Suppresses Sensitizing Effects of Doxazosin to Osimertinib

To confirm whether autophagy mediates the sensitizing effects of doxazosin to osimertinib, we treated cancer cells with 3-MA, an inhibitor of the early step of autophagy [46], in combination with doxazosin and osimertinib. The treatment with 3-MA attenuated the induction of LC3-II by doxazosin and suppressed the sensitizing effects of doxazosin to osimertinib (Figure 4a,b). These results suggest that doxazosin sensitized cancer cells to osimertinib at least partly through the induction of autophagy.

## 4. Discussion

Osimertinib, which is a third-generation EGFR-TKI, is an important anticancer drug because of its high efficacy and safety. However, since the development of resistance to osimertinib is inevitable, therapeutic strategies to overcome resistance are warranted. Doxazosin is an alpha 1-adrenoceptor antagonist used in the treatment of hypertension [19] and acute urinary retention caused by benign prostatic hyperplasia [20]. Regarding drug repositioning or repurposing, doxazosin has been reported to exert anticancer effects by inducing apoptosis and is a candidate drug for cancer [44,45]. However, it currently remains unclear whether it sensitizes cancer cells to osimertinib. In the present study, we showed that doxazosin enhanced the anticancer effects of osimertinib in the cancer cells and CSCs of NSCLC, pancreatic cancer, and glioblastoma cell lines. 

Autophagy is a self-digesting process that results in the breakdown of cellular components by lysosome. Under cellular stress conditions, such as the depletion of nutrients, the activation of autophagy provides an alternative source of energy and enables cells to survive [47]. Autophagy plays a dual role in cancer depending on the type, stage, and context of cancer [48,49,50]. In the early stage of tumorigenesis, autophagy acts as quality control for cells by eliminating damaged cells under carcinogenic stress, thereby suppressing tumor initiation [51]. In the later stages, autophagy is activated, allows cancer cells to survive in an environment depleted of nutrients and oxygen, and then promotes tumorigenesis [52]. Thus, in order to target autophagy for cancer treatment, it is important to consider the role of autophagy in a specific context. In this regard, different roles of autophagy in resistance to EGFR-TKI have been reported [15,16,17,18]. Chloroquine, an inhibitor of autophagy, sensitizes NSCLC cells to EGFR-TKI, indicating a cytoprotective role for autophagy [15,53]. In contrast, the combination of SAHA, a histone deacetylase inhibitor, with EGFR-TKI enhances autophagy-mediated cell death in NSCLC cells [54]. Similarly, curcumin, a compound found in the plant Curcuma longa, induces autophagy and overcomes resistance to EGFR-TKI [55]. Consistent with these latter reports, the present results showed that doxazosin induced autophagy and sensitized cancer cells and CSCs to osimertinib. The sensitizing effects of doxazosin were attenuated by the treatment with 3-MA, an autophagy inhibitor, suggesting that doxazosin reduces resistance to osimertinib by inducing autophagy. In support of the concept that doxazosin sensitizes cancer cells to EGFR-TKI, Hui et al. reported that doxazosin enhanced gefitinib-induced apoptosis in breast cancer cells by inhibiting EGFR through a different mechanism to that described in the present study [25]. Our results suggest that the induction of cytotoxic autophagy by doxazosin is an effective clinical approach to enhance the effects of osimertinib.

Doxazosin is used for patients with cancer to control therapy-related hypertension caused by anti-VEGF drugs [56] and is well-tolerated by these patients. Doxazosin at the concentration used in the present study exerted anticancer effects without affecting the growth of IMR90 non-tumor cells. Although the concentration was at the lower end of the range used in previous studies on cancer cells (10–100 µM) [44,45], it is still above clinically relevant doses, as indicated by Tahamtzopoulos et al. [57]. However, the administration of clinically relevant doses of doxazosin significantly suppressed xenografts of pituitary adenoma and ovarian cancer in mice [58,59]. These findings suggest that doxazosin could be more potent in vivo than in vitro, which is in favor of its clinical application for the treatment of cancer. However, to confirm the clinical applicability, it is necessary to perform further experiments with more clinically relevant models such as three-dimensional (3D) culture and patient-derived xenograft models.

Although the anticancer effects of doxazosin have been reported in several types of cancers, its effects on CSCs remain unknown [23]. CSCs are a small population of cells that have a high tumor initiation capacity; therefore, CSCs are involved in tumorigenesis, recurrence, and metastasis [60]. We showed that doxazosin exerted anti-CSC effects, which raised the possibility that doxazosin could modulate tumorigenesis, recurrence, and metastasis. Patients treated with doxazosin for a long time showed a lower incidence of prostate cancer and bladder cancer, suggesting its tumor preventive role [61,62]. Furthermore, DZ-50, which is a doxazosin derivative, suppressed tumor metastasis in a mouse metastasis model [29]. Although the involvement of the anti-CSC effects of doxazosin was not discussed in these studies, the present results suggest that the suppression of CSCs partially contributes to its preventive effects on tumorigenesis and metastasis.

There are some potential limitations in this study. We examined the combinational effects of doxazosin and osimertinib in cancer cell lines and CSC lines only in vitro. In addition, the clinical applicability of the concentration of doxazosin in our studies remains unknown. Therefore, it is necessary to examine the combinational effects with more clinically relevant models in future studies.

In conclusion, herein, we demonstrated that doxazosin exerted osimertinib-sensitizing effects on various types of cancers via the upregulation of autophagy. Although future studies are needed, the present results clearly demonstrated the potential of doxazosin as an osimertinib sensitizer in many cancer treatment settings.

## 5. Conclusions

Doxazosin overcomes osimertinib resistance through the induction of cytotoxic autophagy in various types of cancer cells and CSCs and is a candidate drug as a sensitizer of osimertinib in many cancer treatment settings.

## Figures and Tables

**Figure 1 biomedicines-08-00273-f001:**
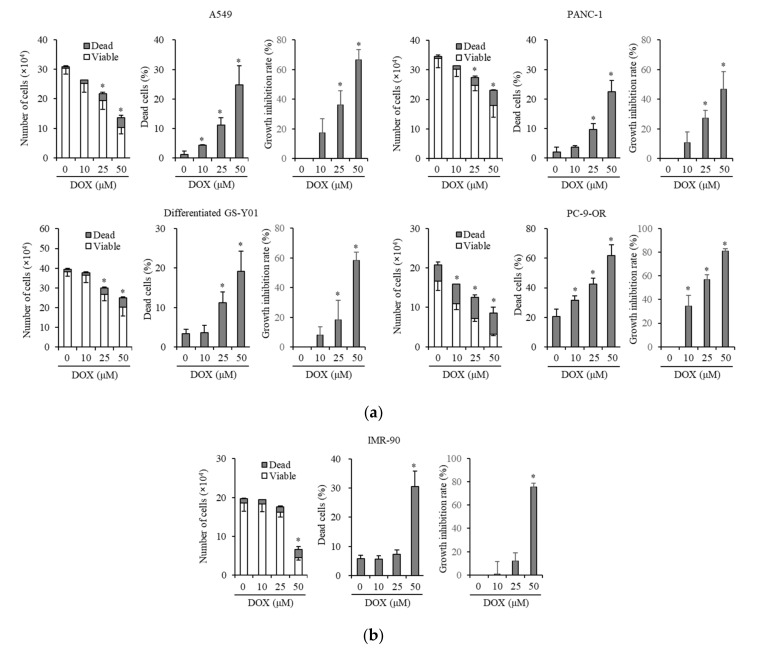
Doxazosin induces cancer cell death and inhibits cancer cell growth without affecting non-tumor cells. A549, PANC-1, differentiated GS-Y01 (differentiated glioma stem cells induced by serum in culture medium), PC-9-OR (subline from PC-9, PC-9 osimertinib resistant) (**a**) and IMR-90 (normal human fibroblasts) cells (**b**) were treated with the indicated concentrations of doxazosin (DOX) for 3 days, and the numbers of viable and dead cells (**left panels**), the percentage of dead cells (**center panels**), and the growth inhibition rate (**right panels**) were assessed. Values in the graphs represent the means ± SD of triplicate samples of a representative experiment repeated with similar results. * p < 0.05.

**Figure 2 biomedicines-08-00273-f002:**
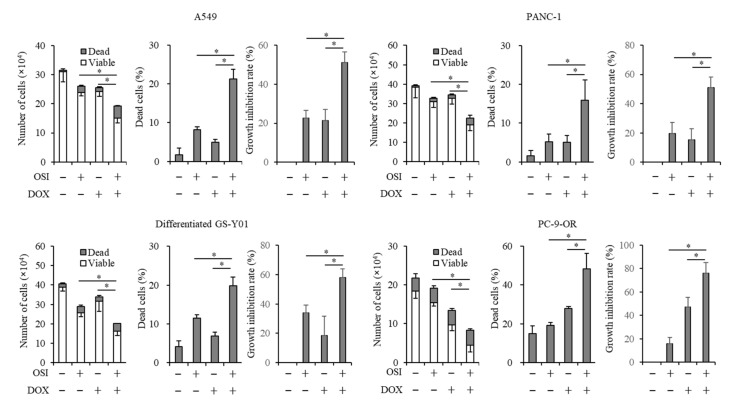
Doxazosin sensitizes several types of cancer cells to osimertinib. Cells were treated with/without 2 µM osimertinib (OSI) with/without 15 µM doxazosin (DOX) for 3 days, and the numbers of viable and dead cells (**left panels**), the percentage of dead cells (**center panels**), and the growth inhibition rate (**right panels**) were assessed. Values represent the means ± SD of triplicate samples of a representative experiment repeated with similar results. * p < 0.05.

**Figure 3 biomedicines-08-00273-f003:**
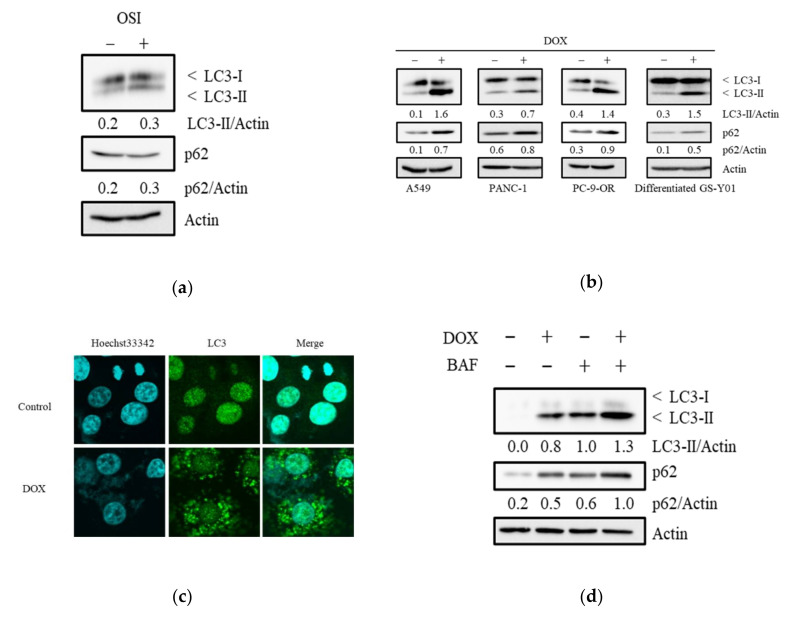
Doxazosin activates autophagy. Cells were treated (**a**) with/without 2 µM osimertinib (OSI) (A549 cells); (**b**) with/without 15 µM doxazosin (DOX); or (**d**) with/without doxazosin in the presence/absence of 10 nM bafilomycin A1 (BAF) (A549 cells), for 3 days, and then subjected to an immunoblot analysis for autophagic markers. A549 cells were treated with/without 15 µM DOX for 3 days, and then subjected to an immunofluorescence analysis (**c**). The relative density was shown below each band of immunoblot.

**Figure 4 biomedicines-08-00273-f004:**
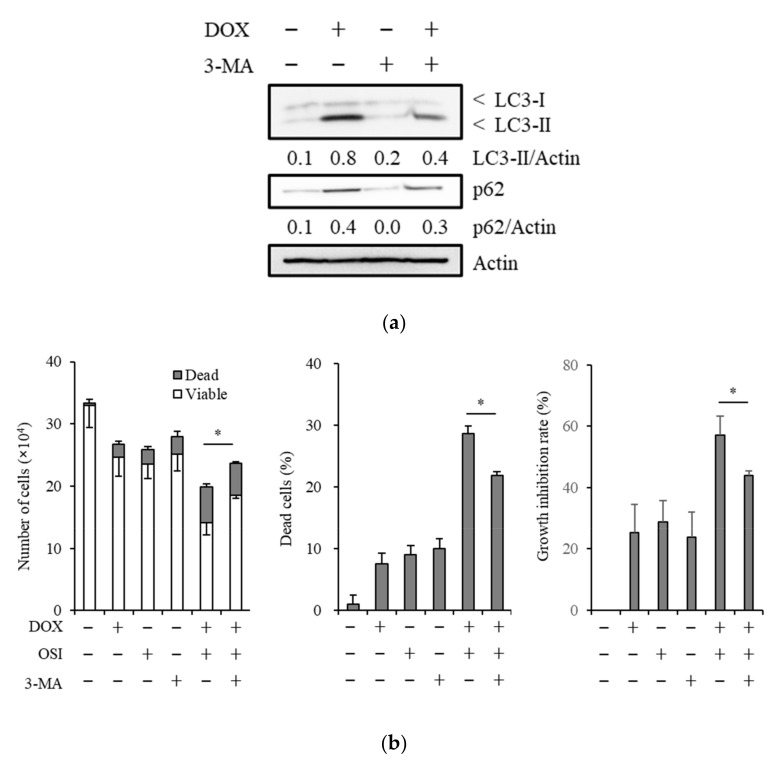
An autophagy inhibitor partially reverses the induction of autophagy and osimertinib-sensitizing effects of doxazosin. (**a**) A549 cells were treated with/without 15 µM doxazosin (DOX) and with/without 0.5 mM 3-methyladenine (3-MA) for 3 days, and then subjected to an immunoblot analysis; (**b**) A549 cells were treated with/without 15 µM DOX, with/without 2 µM osimertinib (OSI), and with/without 0.5 mM 3-MA for 3 days, and the numbers of viable and dead cells (left panel), the percentage of dead cells (center panel), and the growth inhibition rate (right panel) were assessed. Values represent the means ± SD of triplicate samples of a representative experiment repeated with similar results. * p < 0.05. The relative density was shown below each band of immunoblot.

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
