# Peer review of "Doxazosin, a Classic Alpha 1-Adrenoceptor Antagonist, Overcomes Osimertinib Resistance in Cancer Cells via the Upregulation of Autophagy as Drug Repurposing"

_biomedicines, 2020, doi:10.3390/biomedicines8080273_

Round 1

Reviewer 1 Report

The presented manuscript describes the activity of doxazosin in several cancer cell lines. the drug was applied alone or in combination with osimertinib. Especially, the ability of doxazosin to induce autophagy was investigated. The experiments were well planned and the results are sound and of potential importance. The text of the paper is well written. However I have got some remarks:

  1. The statement that doxazosin is non-toxic to normal cells is based on the experiments with only one non-cancerous cell line. Additional normal cell lines should be tested.
  2. Doxazosin was used in combination with osimertinib at the concentration of 2 uM. There is however no information on cytotoxicity of osimertinib alone towards the studied cell lines?
  3. Autophagy activation – the effect of osimertinib alone and doxazosin alone were investigated. And what would be effect of the mixture of both drugs?
  4. Figure legends – it would be better to give time of incubation in hours than in days.
  5. Figure 4 is unclear. It would be advisable to explain the number under the blots in the legend.
  6. Materials and Methods – what were the dilutions of antibodies?
  7. Materials and Methods – please specify the secondary antibody used for immunoblotting and for immunofluorescence experiments.

Author Response

1. The statement that doxazosin is non-toxic to normal cells is based on the experiments with only one non-cancerous cell line. Additional normal cell lines should be tested.

Thank you for your comment. Because there are various types of normal tissues and cells, it is neither practical nor feasible to test the toxicity of a given drug on all of them in vitro. Instead, IMR-90, a normal lung fibroblast cell line we used in our study, is widely and commonly used as a representative non-cancerous cell line to screen drug toxicity in vitro.

2. Doxazosin was used in combination with osimertinib at the concentration of 2 uM. There is however no information on cytotoxicity of osimertinib alone towards the studied cell lines?

We previously reported the cytotoxicity of osimertinib in A549 cells in a dose-response manner (Figure S1, Sanomachi and Suzuki et al., Cancers 2019 11(10):1550). Because 2 µM of osimertinib suppresses viable cell number by approximately 10%–30% in all the studied cell lines and is a commonly used concentration for cancer cells in vitro, we chose the concentration to examine the combinational effects with doxazosin. We inserted the explanation for the reason why we chose the concentrations of osimertinib as well as doxazosin in the text.

3. Autophagy activation – the effect of osimertinib alone and doxazosin alone were investigated. And what would be effect of the mixture of both drugs?

We investigated the combination effects of osimertinib and doxazosin on autophagy. We provided the data in Figure S4 and inserted the description in the Result section of the revised manuscript.

4. Figure legends – it would be better to give time of incubation in hours than in days.

The incubation periods of these experiments were not exactly 72 hours. The description “3 days” is more appropriate than 72 hours as experimental procedures.

5. Figure 4 is unclear. It would be advisable to explain the number under the blots in the legend.

We added the explanation of the numbers in the figure legends.

6. Materials and Methods – what were the dilutions of antibodies?

We added the information about dilution of primary antibodies in Materials and Methods.

7. Materials and Methods – please specify the secondary antibody used for immunoblotting and for immunofluorescence experiments.

We added the information of secondary antibodies in Materials and Methods.

Reviewer 2 Report

In this paper, Suzuki S et al. assessed the effect of doxazocin on the resistance of cancer cells to osimertinib. The authors evoke that doxazocin reverses osimertinib resistance by increasing autophagy and suggest its clinical relevance in combination with osimertinib.

Although the findings are promising, this study suffers from the lack of additional convincing data to support the potential of such combination in clinic. Indeed, I have several concerns regarding the clinical translational potential of this study and the experimental procedure and conclusions:

  • In this study, only conventional 2D cell culture was used. However, 2D cell cultures do not reflect the real structural organization of the cells in vivo which may have considerable impact on the cell death effect of drugs. A more clinically relevant 3D model should be used. In addition, cell lines do not represent the complexity of the genetic and epigenetic landscape of the original tumor and cannot accurately predict response to treatment in clinic. Confirmation of the findings need to be performed in patient derived-cells.
  • The viability assay should be more informative. Although autophagy was suggested as a cell death mechanism in this study, other cell-death mechanisms should be investigated.
  • The drug combination analysis is weak with only one concentration and one time point per drug : A more advanced analysis should be performed following for example the median-effect method with a clear assessment of combination effects (synergism, addition or antagonism).
  • Since no time-course analysis was done to study autophagy, the levels of LC3-II and p62 should be evaluated in the presence of chloroquine that prevents autophagosome–lysosome fusion. Without this condition, no clear conclusion can be raised on the effect of the drugs on the autophagic flux. Also, the authors stated that p62 levels are increased with doxazocin but they don’t explain how this increase is compatible with an activation of autophagy (p62 is degraded by autophagy).
  • The discussion contains hasty conclusions such as: “these findings suggest that doxazocin is more potent in vivo than in vitro and support its clinical application to the treatment of cancer” and “ we showed that doxazosin exerted anti-CSC effects, which implies that it suppresses tumorigenesis, recurrence, and metastasis”.

In conclusion, although this study is a good basis for a deeper investigation of the clinical potential of doxazocin in combination with osimertinib, it is only preliminary and needs to be supported by further data and clinically relevant models.

Author Response

  1. In this study, only conventional 2D cell culture was used. However, 2D cell cultures do not reflect the real structural organization of the cells in vivo which may have considerable impact on the cell death effect of drugs. A more clinically relevant 3D model should be used. In addition, cell lines do not represent the complexity of the genetic and epigenetic landscape of the original tumor and cannot accurately predict response to treatment in clinic. Confirmation of the findings need to be performed in patient derived-cells.

Thank you for your suggestion. In this study, we established the basis of the concept that doxazosin sensitizes cancer cells and cancer stem cells to osimertinib with 2D-cell culture. We agree that we need further studies to make the concept closer to clinical settings using 3D-culture or patient derived-xenograft models, as you indicated. We added the idea to the revised manuscript.

  1. The viability assay should be more informative. Although autophagy was suggested as a cell death mechanism in this study, other cell-death mechanisms should be investigated.

Although our data suggested that autophagy increased by doxazosin sensitized the cancer cells and CSCs to osimertinib, we do not exclude the contributions of other cell-death mechanisms. We have examined the contribution of apoptotic cell death. The results, which are presented in supplementary Fig S4, showed some but less consistent contribution of apoptotic cell death-mediated mechanism than autophagy on doxazosin’s chemosensitization effects.

  1. The drug combination analysis is weak with only one concentration and one time point per drug : A more advanced analysis should be performed following for example the median-effect method with a clear assessment of combination effects (synergism, addition or antagonism).

Thank you for your suggestion. We chose the concentrations of doxazosin and osimertinib to suppress the cell viability by approximately 10%–40% in all the cell lines so that we can examine the chemosensitization effects of doxazosin on osimertinib. I would like to stress the point that doxazosin enhanced the anti-cancer effects of osimertinib regardless of modes of action (synergistic or additional), although the analysis of modes of action is interesting to understand the mechanism more precisely.

  1. Since no time-course analysis was done to study autophagy, the levels of LC3-II and p62 should be evaluated in the presence of chloroquine that prevents autophagosome–lysosome fusion. Without this condition, no clear conclusion can be raised on the effect of the drugs on the autophagic flux. Also, the authors stated that p62 levels are increased with doxazocin but they don’t explain how this increase is compatible with an activation of autophagy (p62 is degraded by autophagy).

Autophagy is a complex cellular process composed of initiation, nucleation, elongation, fusion, and degradation. Our main aim of this study is to reveal the involvement of autophagy in doxazosin’s chemosensitization effect. Because 3-MA acts on the early step of autophagy and suppressed doxazosin-induced autophagy in this study, it suggests that doxazosin acts upstream of the step where 3-MA interferes with autophagy. We performed the experiment with bafilomycin, an inhibitor of fusion of autophagosome and lysosome. Bafilomycin combined with doxazosin increased the level of LC3-II more than that of doxazosin-treated cells. Thease data clearly indicated that doxazosin increased autophagy flux in the cancer cells. Thanks to your advice, we were able to show the effect of doxazosin on autophagy flux. In our study, doxazosine increased p62 protein in cancer cells. Since p62 activation is known to be involved in selective autophagy, we interpreted the results as suggesting the possible involvement of selective autophagy in doxazocin-induced autophagy, though there could of course be other possibilities.

  1. The discussion contains hasty conclusions such as: “these findings suggest that doxazocin is more potent in vivo than in vitro and support its clinical application to the treatment of cancer” and “ we showed that doxazosin exerted anti-CSC effects, which implies that it suppresses tumorigenesis, recurrence, and metastasis”.

According to your comments, we revised the description.

Reviewer 3 Report

Major concern

  1. It is highly recommended that data of cytotoxicity data present as the growth inhibition rate, not percent of dead cells in each test.
  2. Through the results, it is no difference between Figure 1 and 2. The authors should provide evidence of cell line difference used for these two sets of tests instead of mention that is normal cancer cell and CSC.
  3. Several data are not consistent between different tests. For example, the toxicity effect of DOX toward GS-Y01 is similar between figur2 and figure 3, but the dose used is 10 and 15mM, respectively.
  4. In section 2.4, the authors need to demonstrate the autophagy induced by osimertinib at all the cell lines.

Author Response

1. It is highly recommended that data of cytotoxicity data present as the growth inhibition rate, not percent of dead cells in each test.

In supplementary Figures S1, S2, S3, and S5, we provided the data of growth inhibition rate, which help readers better understand.

2. Through the results, it is no difference between Figure 1 and 2. The authors should provide evidence of cell line difference used for these two sets of tests instead of mention that is normal cancer cell and CSC.

We added the description of the difference of the cells in the results section.

3. Several data are not consistent between different tests. For example, the toxicity effect of DOX toward GS-Y01 is similar between figur2 and figure 3, but the dose used is 10 and 15mM, respectively.

According to our results, there was no substantial difference between the effects of 10 µM and 15 µM of DOX on GS-Y01. However, we agree that there were some variations of data between experiments, which might have given the impression that there was inconsistency among data, though in fact there wasn’t.

4. In section 2.4, the authors need to demonstrate the autophagy induced by osimertinib at all the cell lines.

Thank you for your constructive suggestion. It takes two weeks to induce differentiation in GS-Y01, and PC-9OR cells grow very slowly. Because of the limitation of time, we did not perform the experiments with GS-Y01 and PC-9OR. Instead, we performed the experiments with A549, A549 CSLC, PANC-1, and PANC-1 CSLC, which are representative cell lines in this study. The data were provided in Figures S4 of the revised manuscript. The data dramatically improved the quality of this study.

Round 2

Reviewer 1 Report

no further comments

Author Response

Thank you for your positive decision.

Reviewer 2 Report

Despite the efforts done by the authors to strengthen their findings, in my opinion this article does not meet the publication criteria in its revised version. Please find below my comments.

  • I requested a validation of the data in a clinically relevant model (3D cell culture or patient-derived cells). The authors mentioned in the discussion that such experiments are needed but they did not perform any. I still believe that nowadays such experiments are mandatory.

  • “Although osimertinib was previously shown to induce autophagy in NSCLC cells [34,35], the treatment with osimertinib did not induce LC3-II in A549 cells (Figure 4a), PANC-1, A549 CSLC, and PANC-1 CSLC cells (Figure S4)”

The analysis of autophagy was performed after 3 days of treatment, and not at early time-points. As I mentioned in my first report the fact that no clear changes in LC3II levels are visible may simply reflect the degradation of LC3II in these cells because of autophagy activation. That is why I suggested previously to use an autophagy inhibitor before concluding that autophagy is not activated. The authors added a condition with Bafilomycin A1 to show an increase in autophagy in one cell line (Which cell line? It is not mentioned in Figure 4d) due to Dox (also, there is an error in the figure legend where it is stated that the western blot show the results “with/without osimertinib”). However, they did not show that in the presence of Baf A1 there is no osimertinib-induced increase of LC3II in A549 cells, PANC-1, A549 CSLC, and PANC-1 CSLC cells. Therefore, it is not correct to conclude about autophagy in these cell lines without this condition. Also, I have to say that the new data in Suppl Fig 4 tend to show a slight increase of LC3II and a decrease of p62 levels in A549 and A549 CSLC after osimertinib treatment which may suggest au autophagy activation rather than inhibition. Therefore, the conclusions of the authors regarding the autophagy analysis is not very convincing and other controls need to be added before raising any conclusion.

  • In order to evaluate whether apoptotic mechanisms were involved in doxazocin-induced cell death-the authors analyzed the levels of cleaved caspase-3 and concluded that “doxazosin, osimertinib, and their combination did not consistently induce apoptosis in A549, PANC-1, A549 CSLC, and PANC-1 CSLC (Figure S4).”

Even though cleavage of caspase-3 indicates apoptosis activation, one cannot rely only on this effect to conclude about apoptosis especially after 3 days of treatment (here also, as I suggested previously for autophagy analysis, time course experiments would have clearly shown if there is or not induction of apoptosis). Indeed a progressive decrease of the levels of cleaved caspases during apoptosis was already demonstrated in other studies and is probably due to apoptosis itself. Other markers of apoptosis need to be analyzed in the presence or absence of apoptosis inhibitor. Therefore, I do not agree with the authors on the conclusion raised by their analysis.

Author Response

<To comment of Reviewer 2>

  • I requested a validation of the data in a clinically relevant model (3D cell culture or patient-derived cells). The authors mentioned in the discussion that such experiments are needed but they did not perform any. I still believe that nowadays such experiments are mandatory.

We think that clinically-relevant models are necessary for the next step of this project, as mentioned in the discussion. The aim of this study is to confirm the basic concept of the anti-cancer and the chemosensitization effects of doxazosin. We appreciate your suggestion to facilitate the clinical application of our study.

  • “Although osimertinib was previously shown to induce autophagy in NSCLC cells [34,35], the treatment with osimertinib did not induce LC3-II in A549 cells (Figure 4a), PANC-1, A549 CSLC, and PANC-1 CSLC cells (Figure S4)”The analysis of autophagy was performed after 3 days of treatment, and not at early time-points. As I mentioned in my first report the fact that no clear changes in LC3II levels are visible may simply reflect the degradation of LC3II in these cells because of autophagy activation. That is why I suggested previously to use an autophagy inhibitor before concluding that autophagy is not activated. The authors added a condition with Bafilomycin A1 to show an increase in autophagy in one cell line (Which cell line? It is not mentioned in Figure 4d) due to Dox (also, there is an error in the figure legend where it is stated that the western blot show the results “with/without osimertinib”). However, they did not show that in the presence of Baf A1 there is no osimertinib-induced increase of LC3II in A549 cells, PANC-1, A549 CSLC, and PANC-1 CSLC cells. Therefore, it is not correct to conclude about autophagy in these cell lines without this condition. Also, I have to say that the new data in Suppl Fig 4 tend to show a slight increase of LC3II and a decrease of p62 levels in A549 and A549 CSLC after osimertinib treatment which may suggest au autophagy activation rather than inhibition. Therefore, the conclusions of the authors regarding the autophagy analysis is not very convincing and other controls need to be added before raising any conclusion.

Thank you for your comments. As you indicated, the increase of LC3-II by osimertinib was slight in A549 and A549 CSLC and not observed in all the cells we examined. We interpret the effects of osimertinib on the cancer cells and CSCs as slight, if any. We changed the description in the main text according to your suggestion (line 117, page 4). The mistakes in the Figure legends are corrected (lines 131–132, page 5).

  • In order to evaluate whether apoptotic mechanisms were involved in doxazocin-induced cell death-the authors analyzed the levels of cleaved caspase-3 and concluded that “doxazosin, osimertinib, and their combination did not consistently induce apoptosis in A549, PANC-1, A549 CSLC, and PANC-1 CSLC (Figure S4).” Even though cleavage of caspase-3 indicates apoptosis activation, one cannot rely only on this effect to conclude about apoptosis especially after 3 days of treatment (here also, as I suggested previously for autophagy analysis, time course experiments would have clearly shown if there is or not induction of apoptosis). Indeed a progressive decrease of the levels of cleaved caspases during apoptosis was already demonstrated in other studies and is probably due to apoptosis itself. Other markers of apoptosis need to be analyzed in the presence or absence of apoptosis inhibitor. Therefore, I do not agree with the authors on the conclusion raised by their analysis.

Because the cell death induced by these drugs actively occurred at 3 days after the treatments, we chose the time point for the analysis of cleaved caspase-3. However, the elevation of cleaved caspase-3 was slight and inconsistent through the cells. Therefore, we considered that the contribution of apoptotic cell death was not apparent.

Reviewer 3 Report

Major concern

  1. The rationale of this study is to demonstrate the benefit of DOX to enhance the drug sensitivity of cancer cells toward EGFr-TKI. However, through data comparison between figure S1 and S2, seems no difference in drug sensitivity between cancer stem cells and wild type cancer cells. It means this study just demonstrates the add effect of DOX. The authors are suggested to reorganize the study rationale.
  2. Toxicity profile of test drugs in this study preferring using the percentage of the dead cells. However, growth inhibition includes cell death and growth delay. It’s highly recommended using growth inhibition rate instead of dead cell percentage.

Author Response

<To comments of Reviewer 3>

  • The rationale of this study is to demonstrate the benefit of DOX to enhance the drug sensitivity of cancer cells toward EGFr-TKI. However, through data comparison between figure S1 and S2, seems no difference in drug sensitivity between cancer stem cells and wild type cancer cells. It means this study just demonstrates the add effect of DOX. The authors are suggested to reorganize the study rationale.

Thank you for your suggestion. While CSCs are considered to be more resistant to chemotherapeutic reagents than non-CSCs in general, they are not necessarily chemoresistant to all reagents. In this study, osimertinib and doxazosin showed similar effects on cancer cells and CSCs, as you indicated. Therefore, we suggest that doxazosin chemosensitizes osimertinib in cancer cells as well as CSCs. To avoid confusion, we moved the data of the CSCs to the supplementary figures (Figures S1, S2, and S3) and combined the relevant sections (sections 2.1 and 2.2), which were separate in the previous version, into a single section.

  • Toxicity profile of test drugs in this study preferring using the percentage of the dead cells. However, growth inhibition includes cell death and growth delay. It’s highly recommended using growth inhibition rate instead of dead cell percentage.

Thank you for your comment. I agree that growth delay is also an important factor for growth suppression. We moved the data of growth inhibition rate from the supplementary figures to the main figures (Figures 1, 2, and 4).

This manuscript is a resubmission of an earlier submission. The following is a list of the peer review reports and author responses from that submission.